# Redistribution of the SWI/SNF Complex Dictates Coordinated Transcriptional Control over Epithelial–Mesenchymal Transition of Normal Breast Cells through TGF-β Signaling

**DOI:** 10.3390/cells11172633

**Published:** 2022-08-24

**Authors:** Sham Jdeed, Máté Lengyel, Iván P. Uray

**Affiliations:** Department of Clinical Oncology, Faculty of Medicine, University of Debrecen, 4032 Debrecen, Hungary

**Keywords:** bexarotene, ARID1A, SWI/SNF, epithelial–mesenchymal transition, TGF-β, FoxQ1

## Abstract

Therapeutic targets in cancer cells defective for the tumor suppressor ARID1A are fundamentals of synthetic lethal strategies. However, whether modulating ARID1A function in premalignant breast epithelial cells could be exploited to reduce carcinogenic potential remains to be elucidated. In search of chromatin-modulating mechanisms activated by anti-proliferative agents in normal breast epithelial (HME-hTert) cells, we identified a distinct pattern of genome-wide H3K27 histone acetylation marks characteristic for the combined treatment by the cancer preventive rexinoid bexarotene (Bex) and carvedilol (Carv). Among these marks, several enhancers functionally linked to TGF-β signaling were enriched for ARID1A and Brg1, subunits within the SWI/SNF chromatin-remodeling complex. The recruitment of ARID1A and Brg1 was associated with the suppression of TGFBR2, KLF4, and FoxQ1, and the induction of BMP6, while the inverse pattern ensued upon the knock-down of ARID1A. Bex+Carv treatment resulted in fewer cells expressing N-cadherin and dictated a more epithelial phenotype. However, the silencing of ARID1A expression reversed the ability of Bex and Carv to limit epithelial–mesenchymal transition. The nuclear levels of SMAD4, a canonical mediator of TGF-β action, were more effectively suppressed by the combination than by TGF-β. In contrast, TGF-β treatment exceeded the ability of Bex+Carv to lower nuclear FoxQ1 levels and induced markedly higher E-cadherin positivity, indicating a target-selective antagonism of Bex+Carv to TGF-β action. In summary, the chromatin-wide redistribution of ARID1A by Bex and Carv treatment is instrumental in the suppression of genes mediating TGF-β signaling, and, thus, the morphologic reprogramming of normal breast epithelial cells. The concerted engagement of functionally linked targets using low toxicity clinical agents represents an attractive new approach for cancer interception.

## 1. Introduction

Breast cancer is among the most frequently occurring cancers and a leading cause of cancer-related deaths among women, second to lung cancer [1]. Although recent targeted therapeutic options made a significant impact on the outcome at every stage of the disease, these positive trends are attenuated by increasing incidence of breast cancer cases, emphasizing the importance of identifying genetic susceptibility, routine risk assessment, and the availability of viable preventive options [2]. Anti-estrogenic agents were shown to strongly reduce the odds of developing ER-positive breast cancers and proved that risk-based chemoprevention was feasible [3]. However, because of their selectivity, and low patient compliance due to side effects, there is a need for more tolerable agents with relevant mechanistic targets that will prevent breast cancer independently of hormone receptor status [4].

From the perspective of cancer interception, the suppression of mutability and morphologic changes are both critical [5]. A number of phenotypic and genetic predisposing factors for the transformation of normal epithelial cells to malignant precursors have been identified, although few of them represent viable drug targets. Consequently, there is a need for low toxicity options for pharmacologic interventions that reduce the propensity of cells to accrue transforming capabilities [6]. RXR-selective retinoids have been shown to effectively inhibit ER-negative breast cancer formation in transgenic mice [7]. In search of effective treatment modalities at sub-therapeutic doses for rexinoid-based chemoprevention, the combination of the only FDA-approved rexinoid, bexarotene (Bex) with the adrenergic inhibitor carvedilol (Carv) was identified in high throughput drug screens to exhibit anti-proliferative synergy. To elucidate this synergy at the genomic level, one goal of this study was to explore mechanistic targets with the potential to impact cellular transformation events in ‘normal’ breast epithelial cells through chromatin remodeling, rather than directly affecting cell cycle regulators.

In this study, we determined that the frequently mutated tumor suppressor, AT-rich interaction domain 1A (ARID1A), and its standard partner in the SWI/SNF complex, Brg1, are coordinately enriched upon Bex+Carv treatment at gene enhancer sites associated with the maintenance of an epithelial cell phenotype, and suppress a TGF-β-like response in normal breast cells. ARID1A is considered a prognostic marker in breast cancer with a higher level of metastatic incidence in patients with ARID1A mutations [8,9]. While the role of ARID1A in the control of the cell cycle and DNA repair is well documented, its involvement in the regulation of cellular phenotype and functional reprogramming is poorly understood [10,11]. Here, we show that the SWI/SNF complex differentially regulates chromatin accessibility at enhancer regions of several TGF-β-related genes. Furthermore, the deficiency in ARID1A expression reverses the ability of the anti-proliferative combination of Bex and Carv to promote an epithelial phenotype in normal breast epithelial cells.

## 2. Materials and Methods

### 2.1. Cell Culture and Treatment

Human mammary epithelial (HME-hTert) cells were cultured in MEBM medium (Mammary Epithelial Cell Growth Basal Medium) supplemented with 50 μg/mL bovine pituitary extract, 5 μg/mL insulin, 10 ng/mL human recombinant epidermal growth factor, 0.5 μg/mL hydrocortisone, 30 μg/mL gentamicin, and 15 ng/mL amphotericin-B. Cells were tested to be mycoplasma free and maintained in a humidified atmosphere with 5% CO_2_ at 37 °C. Cells were maintained at a low passage number and were seeded in 6-well plates for protein extraction, 24-well plates for generating RNA samples, or in 20 cm diameter plates for chromatin immunoprecipitation assay. Cells were treated or transfected at 30–40% confluence in triplicate for different time points depending on the experimental purpose. Bexarotene and carvedilol were purchased from MedChemExpress and Sigma, respectively. The agents were dissolved in DMSO and ethanol (50/50%) solution and then diluted in media to the final experimental concentration of 100 nM for bexarotene and 1000 nM for carvedilol unless indicated otherwise. The cells were treated at 40% confluence with TGF-β (Biotec, Lot#5180305401) at 5 ng/mL final concentration, diluted in the culture medium.

### 2.2. Western Blotting

HME-hTert cells plated in a 6-well plate and treated for either 6, 24, or 48 h were lysed in 100 ul RIPA lysis buffer (150 mM sodium chloride, 1.0% NP-40, 0.5% sodium deoxycholate, 0.1% SDS, 50 mM Tris pH 8.0) plus protease inhibitor cocktail (1:100). In order to reduce the sample’s viscosity, cell lysates were sonicated over 5 cycles, 30 s on, 30 s off, using a Diagenode Bioruptor^®^ Plus sonicator, Diagenode Inc., Denville, NJ, USA. Protein lysates were centrifuged at 15,000× *g* for 20 min at 4 °C. Total protein concentration was determined using the BCA assay. Protein samples were denatured using a loading buffer containing Beta-mercaptoethanol at 95 °C for 5 min. Thereafter, 25–30 µg of total proteins were loaded and separated in 6% or 10% SDS-polyacrylamide gels to detect proteins at high or low molecular weight, respectively. Proteins were transferred to a polyvinylidene fluoride membrane (PVDF) for 3 h at 0.12 A for high molecular weight proteins (ARID1A), or 20 min at 0.12 A to detect β-actin protein, or for 45 min at 0.12 A to detect proteins of low molecular weight using the 10% gel. The blots were blocked with 5% low-fat milk in TRIS-buffered saline (TBS) buffer for 1 h at room temperature. Blots were probed with primary antibodies (Appendix A) overnight at 4 °C, against ARID1A (1:500), FoxQ1 (1:250), or β-actin (1:1000), followed by washing steps with TRIS-buffered saline plus 0.1% Tween-20 (TBST) buffer. Membranes were incubated with anti-rabbit (Anti-rabbit IgG, Dy Light™ 800, 51515) or anti-mouse (Anti-mouse IgG, Dy Light™ 680, 5470) conjugated secondary antibodies (1:30,000). Blots were developed using an Odyssey^®^ CLx (LI-COR Biosciences, Lincoln, NE, USA) imaging system (LI-COR). Image J software was used to quantify the signal intensity of the detected bands. The relative abundance of a target protein was calculated based on normalization to β-actin protein levels as a housekeeping protein.

### 2.3. Reverse Transcription-Quantitative Polymerase Chain Reaction (RT-qPCR)

Total RNA was isolated using a NucleoSpin RNA isolation kit (MACHEREY-NAGEL, Ref: 740955.50) according to the manufacturer’s instructions. Thereafter, 1 μL RNA was used for the reverse transcription reaction, to generate complementary DNA, through the following conditions: 50 °C for 30 min followed by an enzyme deactivation step at 72 °C for 5 min using reverse primers specific to the target regions. The produced cDNA molecule was amplified through quantitative real-time PCR. Most of the targets’ transcript levels were amplified using TaqMan assay with the following conditions: initial denaturation at 95 °C for 2 min followed by 40 cycles of denaturation at 95 °C for 10 s then annealing and extension at 60 °C for 30 s for each cycle. Some targets’ gene expression was measured using SYBR Green fluorescent dye with the following PCR program: initial denaturation at 95 °C for 5 min followed by 40 cycles of denaturation at 95 °C for 15 s then annealing and extension at 60 °C for 30 s for each cycle. The absolute quantification method was used for data analysis. Target gene transcript levels were normalized to their corresponding β-actin levels within the same sample. Primers and probes were designed using the Primer Express 3.0.1 program ensuring low primer dimer or hairpin formation. Primers were selected after checking their specificity using the NCBI primer-blast tool. Sequences of the qPCR oligos used are shown in Appendix A.

### 2.4. Chromatin Immunoprecipitation

HME-hTert cells plated in 20 cm diameter plates were treated with Veh or Bex+Carv for 6 h. After treatment, cells were fixed using 1% formaldehyde in PBS for 10 min at RT. Formaldehyde activity was quenched using glycine in PBS at 125 mM final concentration for 10 min at RT. Fixed cells were washed twice using ice-cold phosphate-buffered saline. Cells were lysed using lysis buffer (1% Triton X-100, 0.1% SDS, 150 mM NaCl, 1 mM EDTA pH = 8.0, 20 mM Tris pH = 8.0) plus protease inhibitor tablets (Roche, cOmplete Mini, Cat# 11836153001) (40 mL lysis buffer: 3 protease inhibitor tablets). The cell nuclei pellet was collected by centrifugation at 4 °C for 12,000× *g* for 1 min. The cell nuclei were washed three times with the lysis buffer. Chromatin was sheared through the sonication step using the following conditions: 3 × 5 min long cycles with 30 s on and 30 s off using the Diagenode Bioruptor^®^ Plus sonicator. After centrifugation, the supernatant was collected. The input sample was set aside as 10% of the sheared chromatin from each IP. The sheared chromatin was diluted (1:9) in lysis buffer. A clearing step was added to the procedure in order to reduce the background generated from non-specific interactions, including incubation of the diluted chromatin with 1 a mixture of protein A and protein G (1:1) (Dynabeads, Invitrogen, Thermo Fisher Scientific, Waltham, MA, USA, Dynabeads protein A, and protein G) beads coated with IgG for overnight at 4 °C. In parallel blocked beads by 0.5% filter, BSA in PBS were incubated with antibodies against target proteins overnight at 4 °C in lysis buffer solution. The immunoprecipitation step was performed through the incubation of sheared and cleared chromatin with beads coated with antibodies against the protein of interest overnight at 4 °C. The precipitated beads–antibody–protein–DNA complex was then washed. From this step, we included the input samples. The DNA-protein complex was eluted through incubation with elution buffer (0.1 M NaHCO_3_, 1%SDS) for 15 min at 1000 rpm at RT. The DNA-protein interactions were de-cross-linked by incubating the samples with 0.2 M NaCl overnight at 65 °C. The DNA was purified by the addition of 10 µg RNase A (10 µg/µL), 20 ug proteinase K (20 µg/µL), 0.04 M Tris-HCl pH = 7.0, 0.02 M EDTA pH = 8.0, and the mixture was incubated for 2 h at 45 °C on a thermomixer at 1000 rpm. The DNA was performed using Qiagen MinElute PCR purification kit (REF 28004) for ChIP samples to be sequenced or with High Pure PCR Template Preparation Roche Kit in case of ChIP-qPCR. Finally, we measured DNA concentration by the Qubit dsDNA HS assay. Library preparations and sequencing of ChIP samples were performed by the Genomic core facility (University of Debrecen, Debrecen, Hungary), using 10 ng of DNA and Illuminas TruSeq ChIP Sample Preparation. Sequencing was performed using the Illumina NextSeq 500 instrumental mode. Enrichment of specific target regions in DNA samples were tested in qPCR reactions.

### 2.5. ChIP-qPCR

We validated ARID1A binding events identified in ChIP-Seq through ChIP-qPCR. ARID1A, Brg1, and H3K27ac enrichment to target regions were detected. IP and input samples were diluted 3 times with nuclease-free water after DNA purification. Input samples were used as a control; the DNA copy number of a target region in the IP sample was normalized to the DNA copy number in the corresponding input sample. Thereafter, 1 µL DNA from each sample was used in the qPCR reaction using SYBR Green I Master (Roche Diagnostics GmbH) as a fluorescence reporter dye, with the following PCR conditions: initial denaturation at 95 °C for 5 min followed by 40 cycles of denaturation at 95 °C for 15 s then annealing and extension at 60 °C for 60 s. In order to test assay specificity for the studied target, 1 cycle of melting curve program was added for 15 s at 95 °C followed by 1 min at 60 °C, then 10 s at 95 °C at the last step as the signal was recorded. Primer sequences against ARID1A target regions used for ChIP-qPCR are shown in Appendix A.

### 2.6. ChIP-Seq Data Processing

ARID1A ChIP-Seq data (75 bp Single-end reads) were analyzed using a published computational pipeline [12]. Raw reads were aligned to the hg19 reference genome using Burrows–Wheeler Aligner (BWA). Peaks were called using HMCan [13]. Overlap regions were determined using Bedtools with intersectBed command. Annotation was performed using Homer software with annotatePeaks.pl command. H3K27ac ChIP-Seq datasets were analyzed using Galaxy platform (https://usegalaxy.eu/; accessed on 7 September 2020) [14]. After trimming the low-quality reads, raw reads were aligned to the hg19 reference genome using the Bowtie2 tool. Peaks were called using MACS2 software. Tags were normalized to sequencing depth through the bam-coverage function. To visualize the target’s binding events, bigwig files were generated and uploaded into the Integrative Genomics Viewer (IGV) genome browser.

### 2.7. Immunocytochemistry/Immunofluorescence

Cells were seeded on sterilized coverslips placed in 24-well plates, treated at 30–40% confluence, and then fixed with 4% formaldehyde in PBS for 30 min at RT. After washing steps (3 × 5) with cold PBS, cells were incubated with permeabilization solution (0.5% Triton X-100 in PBS) for 40 min at RT. Non-specific bindings were blocked by incubating the cells with blocking buffer (1% BSA/10% normal goat serum/0.3 M glycine in 0.1% PBS-Tween) for 1 h at RT, followed by 3 washing steps with phosphate-buffered saline, 0.1% Tween (PBST). Cells were incubated with primary antibodies overnight at 4 °C (see Appendix A for Primary Antibody information), followed by washing steps (3 × 5) with PBST. Secondary Ab (Alexa fluor conjugated antibodies 1:1000) diluted in 0.1% BSA in PBST, was added to the cells and incubated for 1 h at RT. The primary–secondary antibodies complex was then cross-linked by 4% formaldehyde in PBST for 10 min, followed by a formaldehyde quenching step using Ammonium chloride at 100 mM final concentration for 10 min at RT. DAPI staining was used to stain cell nuclei (1 µg/mL for 15 min at RT. A fluorescent signal was detected using an inverted fluorescent microscope (LEICA DMi8) through LasX software. Data analysis was performed using the CellProfiler 4.2.1 cell image analysis software.

### 2.8. Fluorescent Image Analysis

The fluorescence intensity of target proteins was measured based on cell-by-cell analysis using CellProfiler 4.2.1 software. Images related to one group of samples were uploaded to the program. Primary objects were selected based on a reference dye; in the case of nuclear proteins DAPI stain, and in the case of cytoplasmic proteins Tubulin-Beta was used. Then the integrated intensity for the target protein was calculated in the selected objects. To measure the nuclear translocation of a certain target, we used two reference dyes; one for the nucleus (primary object) and the other for the cytoplasm (secondary object). We measured the nuclear translocation by dividing the target protein’s nuclear-integrated intensity by the cytoplasmic-integrated intensity and generated a ratio.

### 2.9. siRNA Transfection

HME-hTert cells were seeded in a 24-well plate. Transfection was conducted at 30–40% cell confluence for 2 or 3 days. Transfection conditions were optimized to reach ~90% knockdown level using a pool of three ARID1A-specific siRNAs (TriFECTa^®^ RNAi Kit, Integrated DNA Technologies, Coralville, IA, USA) at 20 nM final concentration and Dharmafect 1 transfection reagent (DharmaCon™, Lafayette, CO, USA, cat#T-2001-02) (1:1). For negative control, we transfected the cells with siRNA against Luciferase transcripts, not expressed in mammalian cells, at 20 nM final concentration. The first hours of transfection were performed in an antibiotic-free reduced serum medium (OPTI-MEM, REF 31985-062) in order to reduce the possibility of the antibiotics being taken up by the cells. ARID1A transcript and protein levels were measured in the transfected cells through RT-qPCR or immunostaining, respectively.

### 2.10. Statistical Analysis

Every experiment was performed three times independently. Data analysis was conducted using the GraphPad Prism 6 software. Statistical values are expressed as the mean ± SD or SEM as indicated. Two-tailed Student’s *t*-test with Welch’s correction was used to identify the significance between the two studied groups. Comparisons between more than two groups were performed using a one-way ANOVA test followed by a Tukey post-hoc test. The *p*-value < 0.05 was considered statistically significant.

## 3. Results

The chemopreventive ability of synthetic RXR-selective retinoids may be potentiated by diverse groups of agents [15,16]. As we showed previously in MCF-7 cells, the adrenergic inhibitor carvedilol (Carv) augmented the growth-suppressive effect of bexarotene (Bex) and suppressed IGF-1 signaling [17]. Because the inhibition of breast cancer formation is generally associated with an antiproliferative effect in normal or premalignant mammary epithelial cells, the interaction of bexarotene and carvedilol was tested in normal mammary epithelial cells immortalized with telomerase (HME-hTert). Cell counts over time revealed a superior efficacy of the combination over bexarotene, while carvedilol alone had marginal effects on growth (Figure 1A). To identify transformation-suppressing mechanisms specific for the drug interaction of Bex and Carv, next RPPA was used to compare expression levels of proteins whose levels only changed upon combined administration of both agents to HME-hTert. In addition, factors required for DNA repair, the maintenance of genomic integrity, and chromatin remodeling were filtered. ARID1A and the ataxia teleangiectasia mutated (ATM) protein matched these criteria [18,19], and displayed 50% (*p* < 0.005), and 2-fold (*p* < 0.05) inductions by Bex+Carv, respectively (Figure 1B). A strong trend of elevated ATM phosphorylation at S1981 was also seen. As we previously identified the SWI/SNF complex as a key growth regulator in breast cancer cells, the induction of ARID1A in HME-hTert cells was confirmed by Western blots and the analysis of immunofluorescent staining (Figure 1C,D).

To clarify the nature of the responses to Bex+Carv in HME-hTert cells that may be mediated by chromatin remodeling events, we compared the genome-wide changes in the acetylation of the histone H3K27 after chromatin immunoprecipitation (ChIP). A functional mapping of the aligned DNA fragments enriched upon Bex+Carv treatment pointed to several genes frequently involved in tumorigenic transformation, the epithelial–mesenchymal transition (EMT) process, and TGF-b signaling (Figure 2A). The altered acetylation state of H3K27 in the enhancers of FoxQ1, KLF4, BMP6, and the TGFBR2 genes were confirmed by quantitative PCR detection of repeated ChIP experiments and were associated with significant changes in the occupancy of one or both key components of the SWI/SNF chromatin-remodeling complex, ARID1A and Brg1, at the same loci (Figure 2B). Brg1 is a standard component of SWI/SNF and was shown earlier to be modulated by ATM-dependent phosphorylation during DNA damage repair [20,21]. Interestingly, the occupancy of ARID1A and BRrg1 at enhancers of FoxQ1, KLF4, TGFBR2, and BMP6 upon exposure to Bex+Carv was increased after just 6 h of drug treatment. In contrast to long-term treatment with Bex+Carv, ARID1A levels remain unchanged after 6 h (Appendix A), therefore, these changes in genomic occupancy suggest a redistribution of SWI/SNF to the affected regions. Furthermore, the enrichment of ARID1A and Brg1 was associated with subsequent suppression of FoxQ1, KLF4, and TGFBR2 mRNA levels and transcriptional activation of the BMP6 gene (Figure 2C). Interestingly, only 8% (18 out of 225) of genes affected by altered acetylation marks and ARID1A occupancy demonstrated significant changes in transcript levels reflected in RNA-seq data (Appendix A).

A key role of ARID1A in the transcriptional regulation of these genes was further supported by the inverse change of their mRNA levels upon siRNA-mediated knock-down of ARID1A (Figure 2D). Each of these putative ARID1A-dependent target genes can be linked to the morphogenic transition of cells associated with TGF-β signaling [22]. Coincidently, the transcript levels of typical EMT marker genes including Fibronectin 1 and N-cadherin were suppressed by Bex+Carv treatment, while the expression of the epithelial marker E-cadherin increased (Figure 2E), suggesting that this Bex+Carv-induced pattern of SWI/SNF distribution may play a role in the control of the transition between epithelial and mesenchymal states in normal breast cells. In addition, a loss of the spindle shape and a more cobblestone-like phenotype of the cells was observed by fluorescent microscopy, in response to Bex+Carv (Figure 2F).

The suppressed transcription of FoxQ1 resulted in the down-regulation of the Forkhead Box Q1 (FoxQ1) protein, as was demonstrated by Western blots and immunofluorescent staining of HME-hTert cells (Figure 3A,B). Simultaneous labeling of ARID1A and FoxQ1 in HME-hTert cells transfected with non-targeting (siNT) and ARID1A-specific siRNA pools suggested that cells deficient for ARID1A express higher levels of FoxQ1 protein. Cell-by-cell analysis of fluorescent microscopic images indicated that knock-down of ARID1A resulted in low ARID1A expression in a dominant fraction (92% vs. 28.3% in the control) of the cells, while the ratio of high FoxQ1 expressing cells increased from 17.3% to 39% (Figure 3C). Within this population, the ratio of low ARID1A/high FoxQ1 cells changed from 0.3% in control knock-down (siNT) to 34% in siARID1A transfected samples, while the proportion of high ARID1A/low FoxQ1 cells decreased from 54.7% to 3%.

To examine the impact of FoxQ1 on the expression of genes dictating morphologic phenotype, we next determined whether E-cadherin expression was induced upon the addition of Bex+Carv or TGF-β [23]. Using double staining, cell surface E-cadherin expression was detected and correlated with the fraction of FoxQ1 in the nucleus, the location at which the activity of this transcription factor is expected (Figure 3D). Single-cell classification along these parameters identified the fraction of cells with decreased nuclear levels for FoxQ1 and elevated E-cadherin staining to be increased from 4.6% to 18% or 46.5% when treated with Bex+Carv or TGF-β, respectively (Figure 3E).

Another important regulator of TGF-β response is SMAD4 and its translocation to the nucleus when activated [24]. Bex+Carv treatment reduced the total amount of the SMAD4 protein (Figure 4A), along with the fraction localized to the nucleus, versus the cytoplasm, as measured by segmented image analysis of immunofluorescent staining (left segments of Figure 4B,C). However, knock-down of ARID1A by siRNAs reversed this trend and caused Bex+Carv to strongly induce SMAD4 levels and its nuclear translocation (Figure 4B,C). In addition, the levels of nuclear phospho-SMAD2, an activation partner of SMAD4, were also suppressed upon knock-down of ARID1A (data not shown). In single cell analysis the fraction of cells with a high nuclear SMAD4 ratio significantly decreased upon Bex+Carv (13.7% vs. 27.7% in vehicle-treated), and to a lesser extent (19.6%) after TGF-β treatment (Figure 4D,E).

Lower nuclear SMAD4 abundance was associated with a marked decrease in N-cadherin expressing cells. However, an increased fraction of cells expressed higher amounts of N-cadherin when treated with TGF-β. This indicates that SMAD4-mediated regulation of EMT markers may differ from FoxQ1-directed changes. Nevertheless, the suppression of ARID1A had an impact on the response to Bex+Carv by all 3 markers in HME-hTert cells. While fibronectin 1 and N-cadherin levels decreased upon Bex+Carv treatment, prior transfection with ARID1A siRNA caused a marked increase in these mesenchymal markers (Figure 5A,B). In contrast, ARID1A knock-down did not reverse the overall increase in E-cadherin staining by Bex+Carv (Figure 5C).

## 4. Discussion

RXR-selective synthetic retinoids have been known for their potential to suppress breast cancer formation with moderate toxicity [4,25,26]. Although high-affinity binding of RXR agonists allows selective gene regulation leading to growth control, for this study we hypothesized that the modulation of chromatin structure may result in mechanistically coordinated changes suppressing the transformation of mammary epithelial cells.

Reducing the toxicity of retinoids for tolerability in the therapeutic, and more so in the chemopreventive setting, has been at the forefront of interest, inspiring the development of many new analogues [27,28]. The use of bexarotene (Bex) and carvedilol (Carv) in combination, both FDA-approved clinical drugs, is beneficial, because the synergy between the two agents may allow their chronic administration at sub-therapeutic doses in order to achieve optimal efficacy with limited toxicity [29]. In a previous study, we characterized the tumor suppressor protein ARID1A as a factor responsible for mediating the anti-proliferative effect of the combination of Bex and Carv, through the suppression of IGF-I-dependent mechanisms in MCF-7 cells [17]. Here, we showed that in normal breast cells the combination of Bex+Carv down-regulated genes and activities (i.e., FoxQ1 and Smad4) that altogether mediate TGF-β signaling.

The concerted action of Bex+Carv to suppress several regulators of the TGF-β signaling pathway is novel and in accordance with the pharmacologic profile of rexinoids. A recent study uncovered a direct impact of RXRα in the TGF-β-dependent regulation of the CCL2 gene in triple-negative breast cancer cells, indicating that physical interaction might occur between these proteins [30]. Thus far, the chemopreventive activity of retinoid X receptor-selective agonists has been interpreted mainly in the context of cell cycle control and mitogenic signaling [31,32]. Our data revealed a coordinated response regulating a switch in the expression of the genes augmenting epithelial characteristics over mesenchymal traits in non-malignant cells. This response appeared to be inhibitory, but not uniformly inverse to the actions of TGF-β in immortalized breast epithelial cells. Rexinoids are unequivocally linked to tumor-suppressing activity, whereas TGF-β has been proposed to behave in a dichotomous manner, depending on the state of malignant transformation of the cell [33]. While in the context of malignant transformation TGF-β may be a promoter of cell proliferation and invasion, in normal epithelium TGF-β signaling may convey tumor-suppressive effects by inducing cell cycle arrest, apoptosis, and cellular senescence and autophagy [34]. Primary breast epithelial cells immortalized by expressing telomerase (hTert) were assumed to have lost their responsiveness to TGF-β [35]. Our results demonstrated that HME-hTert cells retained a response of EMT markers to TGF-β, although HME-hTert cells display a largely mesenchymal phenotype in their subconfluent, proliferative state (see Figure 2F, left panel). Remarkably, along with suppressing proliferation, Bex+Carv triggered a concerted response in the regulation of genes critical for TGF-β signaling, which translated into the suppression of the EMT process.

The genome-wide ChIP-seq analysis of altered H3K27 acetylation patterns revealed comprehensive gene networks activated or suppressed by a short-term treatment of normal breast cells with Bex+Carv. Quantitative PCR validation of the enrichment of ARID1A and Brg1 and prior RNA-seq experiments identified several genes linked to the TGF-β pathway, including FoxQ1, KLF4, TGFBR2, and BMP6, that were affected by the recruitment of the SWI/SNF complex and whose expressions changed in the process. While none of the featured ARID1A-dependent genes are frequently altered in advanced invasive or metastatic breast cancers, the responses detected in immortalized but untransformed epithelial cells suggest that Bex+Carv may suppress a pro-carcinogenic signal and the acquisition of invasive capabilities.

TGF-β and BMPs share a similar signaling process, activating downstream transcription factors, the SMAD proteins, upon binding respective ligands to their membrane kinase receptors. However, bone morphogenic proteins (BMPs) often act as negative regulators of TGF-β action [36]. In the context of breast cancer, BMP6, specifically, was shown to inhibit estrogen-dependent cell growth and induce E-cadherin expression in breast cancer cells [37,38]. Furthermore, high expression of BMP6 was associated with greater infiltration of tumors by immune cells with cytolytic activity, and better prognosis in ER-positive breast cancers [39]. Thus, the induction of BMP6 by Bex+Carv is in line with the therapeutic spectrum of these agents.

Kruppel-like factor 4 (KLF4) and FoxQ1 emerged as targets clearly down-regulated the recruitment of SWI/SNF upon Bex+Carv treatment. KLF4 was shown to form the link between TGF-β1-induced gene transcription and H3 acetylation in vascular smooth muscle cells and mediates TGFβ/BMP signaling through the production of BMP6 [40,41]. KLF4 is also required for the maintenance of breast cancer stem cells and for cell migration and invasion [42]. Forkhead Box Q1 (FoxQ1) is a pro-tumorigenic factor whose gene shows subtype-specific differences in expression levels in breast cancers [43]. FoxQ1 was also considered to play a role in the activation of Wnt signaling in solid tumors [44]. However, the suppression of FoxQ1 inhibits invasion and metastasis through the reversal of epithelial–mesenchymal transition in bladder cancer [45]. Furthermore, FoxQ1 was shown to suppress the expression of the key EMT regulator E-cadherin by binding to the E-box in its promoter region. Mechanistic studies have shown that FoxQ1 expression is regulated by TGF-β1 and that silencing of FoxQ1 can reverse TGF-β1-induced EMT at both morphological and molecular levels [23].

Although both transcription factors, FoxQ1 and KLF4, are known mediators of TGF-β signaling, their relationship to the canonical pathways represented by the action of SMADs may be cell-type and context-dependent. Specifically, the cadherins E and N were differentially regulated by TGF-β and Bex+Carv in their response to FoxQ1 or SMAD4, respectively. TGF-β increases the ratio of cells highly expressing N-cadherin, while more of those cells exhibit low nuclear levels of SMAD4. Contrary to TGF-β, Bex+Carv suppresses N-cadherin expression along with lower nuclear SMAD4 levels. On the other hand, while both Bex+Carv and TGF-β appear to inhibit the nuclear accumulation of FoxQ1, both increase the fraction of high E-cadherin expresser cells. Nevertheless, these differences indicate that the effects of Bex+Carv on the transcription of phenotype-shaping genes only partially overlap with the actions of TGF-β. Furthermore, a differential response may result from the relative contribution of canonical and non-canonical pathways of TGF-β signaling, providing a possible explanation for treatment-specific responses [34].

E-cadherin was primarily used in this study as an epithelial phenotypic marker. Nevertheless, E-cadherin exerts a tumor-suppressing role by sequestering β-catenin from the Lymphoid enhancer factor (LEF) and blocking transcription of genes of the proliferative Wnt signaling pathway. The loss of E-cadherin function has been associated with poor prognosis and survival in patients of various cancers. As E-cadherin expression has been implicated with the cellular functions of invasiveness reduction, growth inhibition, apoptosis, cell cycle arrest, and differentiation, it is not entirely clear whether the loss of E-cadherin is the cause or effect of epithelial–mesenchymal transition [46].

One key finding of this study is the coordinated regulation of gene expression through the SWI/SNF chromatin-remodeling complex, pointing to a suppression of TGF-β function and EMT markers. While uncovering a functionally coherent set of genes by SWI/SNF, our research did not address the question of shared or distinct identity of transcription factors involved in the expression of the affected genes. Furthermore, any potential preferences of the SWI/SNF machinery for pathway-specific enhancer sequences remain unclear. In a groundbreaking report establishing a link between SWI/SNF function and TGF-β signaling, Ross et al. proposed that by being recruited to Smad2-dependent promoters, the SWI/SNF ATPase Brg1 mediates the TGF-β-dependent transcription of Smad2 target genes [47]. Confirming this functional connection, our data, in turn, suggests a model in which the recruitment of SWI/SNF to select enhancer regions may limit or enhance the accessibility of transcription factors.

The prospect of a chromatin-level intervention in the phenotype-switching mechanisms using the combination of two low-toxicity agents may represent a novel approach in non-malignant breast epithelial cells. The mechanism by which Bex+Carv may alter chromatin structure in the select areas through the SWI/SNF complex remains hitherto unclear. Nevertheless, while ARID1A was selectively induced by the drug combination, the enrichment of SWI/SNF at their genomic targets was detected prior to the upregulation of ARID1A, implicating the redistribution of the complex as the driver of this change. However, under circumstances of long-term exposure to the drugs, such as during a chemopreventive treatment, the elevated expression and activity of ARID1A may be expected to replicate the effects shown here. Further beneficial changes due to increased activity of ARID1A as a tumor suppressor, including improved DNA damage repair, could be among the expected beneficial outcomes of Bex+Carv treatment.

The loss of ARID1A, a frequently occurring alteration in malignant cells, was shown to result in the loss of the tumor suppressive function of TGF-β and increased cell migration [48]. Our gene silencing results demonstrated that ARID1A deficiency releases transcriptional regulators of TGF-β signaling, FoxQ1, and KLF4, increasing the propensity to cellular transformation and morphologic changes promoting tumorigenesis and dissemination. Knock-down of ARID1A reversed the effect of Bex+Carv on the expression of fibronectin and N-cadherin, leading to increased levels of these mesenchymal markers. Thus, the clear distinction of pharmacological interventions targeting ARID1A may starkly differ in the prevention setting, compared to the context of cancer therapy in the face of ARID1A deficiency. Furthermore, in the course of carcinogenesis the loss of the tumor suppressor ARID1A may predispose early cancer cells to genomic instability. Thus, in normal breast epithelium, the ability to increase ARID1A levels and increased activity of the SWI/SNF complex may be beneficial for the suppression of cancer cell formation.

ARID1A, SWI/SNF activity, and the ATM kinase appeared co-regulated by Bex and Carv in normal breast cells. ATM and ATR kinases are recruited to DNA breaks and phosphorylate Chk1/2, and, thus, may contribute to cell cycle checkpoint arrest. A functional tie between ARID1A and the ATM/Chk2 DNA damage checkpoint axis was also made evident by Wang et al. and suggested an opportunity to potentiate the therapeutic efficacy of immune checkpoint therapy in ARID1A-deficient tumors [49]. A number of synthetically lethal therapeutic opportunities arise on the basis of ARID1A mutations and mismatch repair deficiency, utilizing inhibitors of PARP1, HDAC2/6, Aurora kinase A, CKD4/6, or PI3K/Akt [50]. However, as a novel example of interacting pathways, in our experimental system, the suppression of ARID1A altered the ability of the anti-proliferative agents Bex and Carv to promote an epithelial phenotype and suppress EMT markers in normal breast cells. It has been previously proposed that ARID1A may play a relevant role both in tumor initiation and tumor suppression, depending on the context [51]. In our model system, the genomic redistribution and activation of the SWI/SNF complex and ARID1A by Bex+Carv, and arguably a subsequent up-regulation of the factor, supports the maintenance of epithelial phenotypic traits and prevents normal breast cells from undergoing a mesenchymal transition. In conclusion, the central role of ARID1A in the maintenance of epithelial cell identity indicates a strong potential for the therapeutic targeting of epigenetic mechanisms and chromatin accessibility for the reduction of cancer risk and tumorigenic transformation. Furthermore, we conclude that the comprehensive cellular response resulting in the cancer-preventive activity of rexinoids and synergistic drug combinations may be better explained by the modulation of chromatin structure affecting functionally linked gene groups, rather than interpreted as the sum of individually regulated tumor suppressors.

## Figures and Tables

**Figure 1 cells-11-02633-f001:**
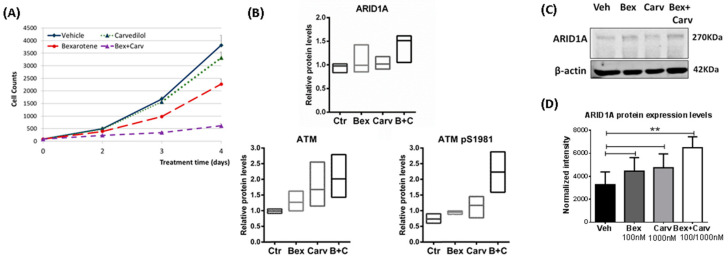
The effect of the combined treatment by Bex and Carv in HME-hTert on cell proliferation and the chromatin surveillance factors ATM and ARID1A. Bexarotene and carvedilol combination treatment showed anti-proliferative effects in normal HME-hTert cells associated with an induction in ARID1A protein levels. (**A**) Dose-response profile of the effect of Bex+Carv combination treatment on HME-hTert cells proliferation treated with the individual or combination treatment for 4 days. (**B**) Reverse phase proteomic array results show the protein levels of ARID1A and ATM as well as the phosphorylated ATM at S1981 upon Bex and/or Carv combination treatment in HME-hTert cells. (**C**) Western blot analysis of ARID1A protein expression levels in extracts from HME-hTert cells treated with the combination of 100 nmole/L Bex and/or 1000 nmole/L Carv vs. vehicle for 48 h. β-actin protein levels were used for normalization. (**D**) Quantification of ARID1A integrated intensity derived from HME-hTert cells treated with the combination or the individual treatments and immunostained with ARID1A. About 200 cells per replicate were analyzed using ImageJ software based on integrated cellular pixel intensities of the immunostaining for ARID1A, normalized to cell numbers based on DAPI nuclear staining. The results are expressed as mean ± SD ** *p* < 0.01 by One-way ANOVA, Tukey post-hoc test.

**Figure 2 cells-11-02633-f002:**
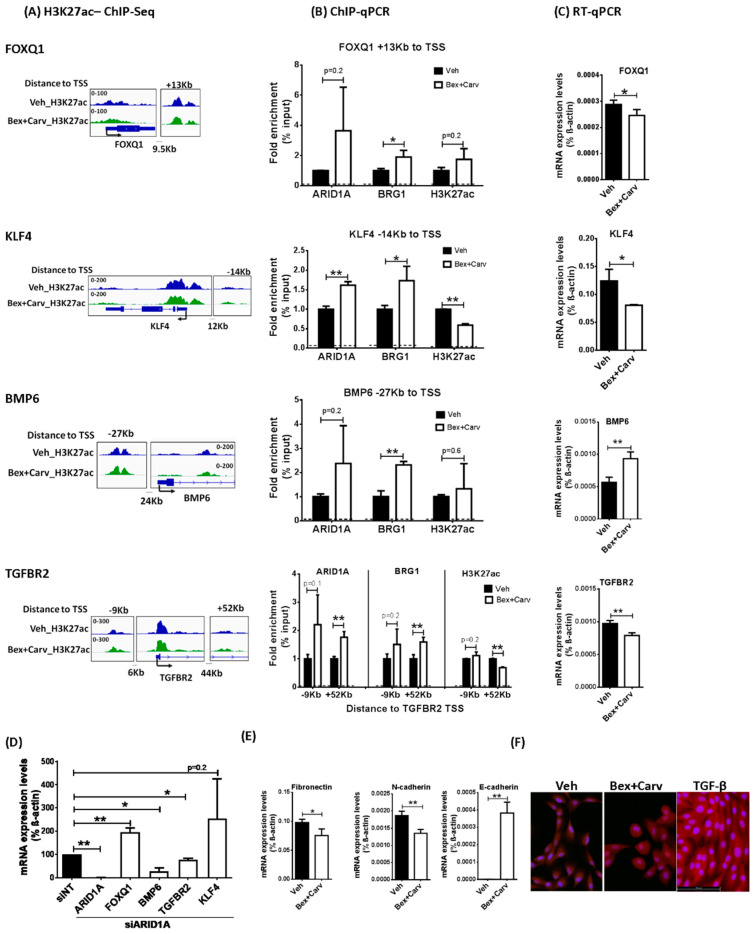
Recruitment of ARID1A to regulatory elements of TGF-β signaling pathway genes upon Bex+Carv treatment and associated change in target gene expression and phenotype in HME-hTert cells. (**A**) ChIP-Seq tracks of H3K27ac in control and Bex+Carv-treated HME-hTert cells across FoxQ1, KLF4, BMP6, and TGFBR2 genes showing the promoter and the studied regulatory regions, exported from the Integrative Genomics Viewer (IGV) application. (**B**) Quantitation of ARID1A and Brg1 enrichment and H3K27ac marks to the putative regulatory elements assigned to FoxQ1, KLF4, BMP6, and TGFBR2 genes after Veh or 100 nmole/L Bex + 1000 nmole/L Carv treatment for 6 h in HME-hTert cells. N = 3 representing 3 biological replicates. The dashed line represents the signal derived from the negative control sample. (**C**) RT-qPCR analysis of target gene transcript levels upon Vehicle or 100 nmole/L Bex + 1000 nmole/L Carv treatment for 6 h in case of KLF4 and BMP6 and 24 h in case of FoxQ1 and TGFBR2. N = 3 representing 3 biological replicates. (**D**) RT-qPCR analysis of ARID1A, FoxQ1, BMP6, and KLF4 gene expression upon 20 nmole/L siRNA transfection for 48 h, and TGFBR2 gene expression upon 72 h of 20 nmole/L siRNA transfection. The gene expression results are presented as a percentage relative to the control sample (siNT). N = 3 representing 3 biological replicates. (**E**) RT-qPCR analysis for the transcript levels of Fibronectin 1 after 24 h, N-cadherin after 6 h, and E-cadherin after 24 h of 100 nmole/L Bex+ 1000 nmole/L Carv combination treatment in HME-hTert cells. The results are expressed as mean ± SD * *p* < 0.05, ** *p* < 0.01 by two-tailed Student’s *t*-test. (**F**) Representative fluorescent images of HME-hTert cell morphology after 72 h of vehicle, Bex+Carv, or TGF-β treatment. Blue: DAPI nuclear stain, Red: CellMask cytoplasmic stain; images taken at 20× magnifications.

**Figure 3 cells-11-02633-f003:**
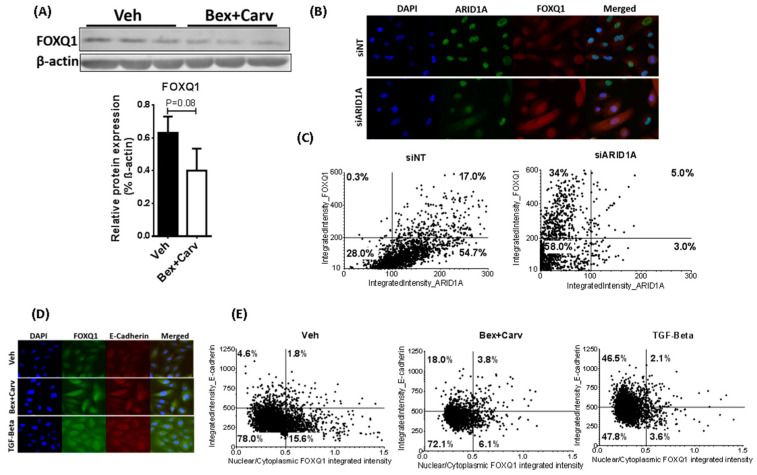
ARID1A-dependent regulation of FoxQ1 and E-cadherin by Bex+Carv and TGF-β in HME-hTert cells. (**A**) (Top) Western blot analysis of FoxQ1 protein levels in HME-hTert cell extracts upon Vehicle vs. 100 nmole/L Bex + 1000 nmole/L Carv treatment for 48 h. (Bottom) quantification of FoxQ1 protein expression levels upon the combination treatment. β-actin was used as a housekeeping gene for normalization. N = 3 representing 3 biological replicates. (**B**) Representative images of HME-hTert cells transfected with siRNA against luciferase as a non-targeting control (siNT) or ARID1A (siARID1A) for 72 h and immunostained against ARID1A (green) and FoxQ1 (red), images were taken at 20× magnification. (**C**) Scatter plots categorizing cells based on their ARID1A and FoxQ1 protein levels upon transfection with siRNA against Luciferase (siNT, left) or against ARID1A (siARID1A, right). A minimum of 1000 cells were individually analyzed using CellProfiler software based on the integrated intensity of ARID1A and FoxQ1 staining in the nucleus or whole cell, respectively. (**D**) Representative images of cells treated with Veh, 100 nmole/L Bex+ 1000 nmole/L Carv for 3 days, or 5 ng/mL TGF-β for 1 day and immunostained with for FoxQ1 and E-cadherin (20× magnification). (**E**) Correlation between FoxQ1 nuclear translocation and E-cadherin levels based on nuclear/cytoplasmic ratio of FoxQ1 and cellular E-cadherin staining in cells treated with Veh (Left), 100 nmole/L Bex+ 1000 nmole/L Carv for 3 days (Middle) or 5 ng/mL TGF-β for 1 day (Right).

**Figure 4 cells-11-02633-f004:**
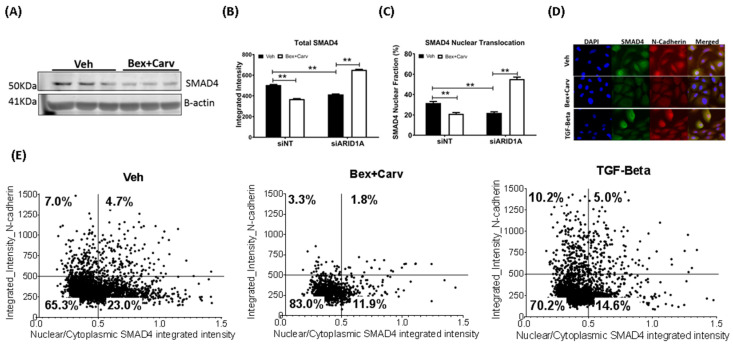
The impact of Bex+Carv or TGF-β treatment on SMAD4 protein expression and nuclear translocation and N-cadherin. (**A**) Western blot analysis for SMAD4 protein levels in HME-hTert cell extracts upon Vehicle vs. 100 nmole/L Bex + 1000 nmole/L Carv treatment for 48 h. (**B**,**C**) Quantification of SMAD4 protein expression (**B**) and its nuclear fraction (**C**) after siRNA-mediated silencing against Luciferase (siNT) or ARID1A (siARID1A) for 72 h and Veh vs. Bex (100 nmole/L) + Carv (1000 nmole/L) treatment for 24 h. A minimum of 500 cells were analyzed individually using CellProfiler for the integrated intensity of SMAD4 in the nucleus and the cytoplasm. The results are expressed as mean ± SEM, ** *p* < 0.01 by One-way ANOVA, Tukey post-hoc test. (**D**) Representative images of cells treated with Veh or Bex+Carv for 3 days, or TGF-β for 1 day and immunostained for SMAD4 and N-Cadherin (20× magnification). (**E**) Scatter plots of nuclear/cytoplasmic ratio of SMAD4 and N-cadherin levels in cells treated with Veh, 100 nmole/L Bex+ 1000 nmole/L Carv for 3 days, or 5 ng/mL TGF-β for 1 day.

**Figure 5 cells-11-02633-f005:**
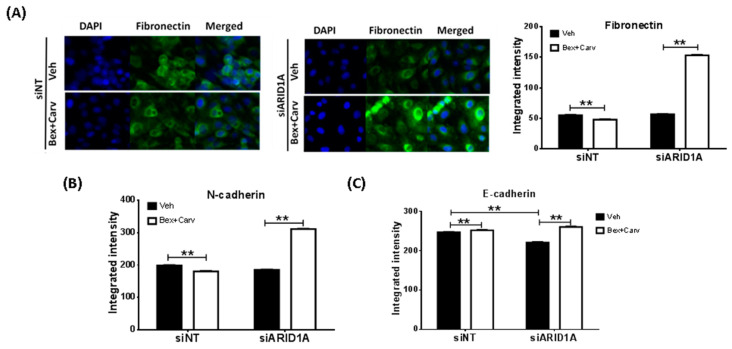
Comparison of the responses to Bex+Carv in EMT marker expression in control and ARID1A-silenced HME-hTert cells. (**A**) Representative images and quantification of cells transfected with siRNA against luciferase as a control (siNT) or ARID1A (siARID1A) for 72 h and treated with Veh or Bex (100 nmole/L) and Carv (1000 nmole/L) for 24 h and immunostained for Fibronectin-1. (**B**,**C**) Quantification of N-cadherin (**B**) and E-cadherin (**C**) protein expression levels upon siNT or siARID1A transfection for 72 h and 24 h of Veh or 100 nmole/L Bex+ 1000 nmole/L Carv treatment. A minimum of 500 cells were analyzed individually using CellProfiler for whole cell integrated intensity of Fibronectin, N-cadherin, or E-cadherin (20× magnification). The results are expressed as mean ± SEM ** *p* < 0.01 by One-way ANOVA, Tukey post-hoc test.

## Data Availability

The sequencing data used in our manuscript are publicly available on NCBI’s Sequence Read Archive (SRA) site under the sample accession numbers: SRX14284269, SRX14284268, SRX14284267, SRX14284266, SRX14284265, SRX14284264, SRX14284263, SRX14284262, BioProject PRJNA810076 (https://www.ncbi.nlm.nih.gov/bioproject/PRJNA810076/; accessed on 17 August 2022).

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
