# Peer review of "Redistribution of the SWI/SNF Complex Dictates Coordinated Transcriptional Control over Epithelial–Mesenchymal Transition of Normal Breast Cells through TGF-β Signaling"

_cells, 2022, doi:10.3390/cells11172633_

Round 1

Reviewer 1 Report

This manuscript was well written and I would suggest this for publication with the following comments answered.

Please see my review below.

Sham Jdeed et al., have performed their study based on TGF beta signaling, and its impact on the EMT process. The study also highlights the mechanism of cancer preventive drugs and the distribution of SWI/SNF complex behind drug treatment. 

The study shows ARID1A orchestration by rexinoid bexarotene and Carvedilol drugs and its effect on EMT-related genes. The following comments might allow the reader to get deeper scientific understanding of the mechanism.

1) From ChIP-seq data it would be nice to show the effect of drugs on genes in cis- and trans- regions specifically related to TGF beta signaling.

2) comparison of DNA binding and mRNA levels is a nice way to present cis-regulation of transcription factors. 

3)Smad4 is shown as an important piece in this mechanism: It was shown that Smad4 levels are disturbed by this mechanism. Smad4 is also known to shift Smad2 and Smad3 complex to nuclear. It would be nice to see if their translocation is also affected.

4) Inturn if, Smad-dependent TGF beta signaling is affected, is there any evidence of a compensation mechanism?

Author Response

Please also see the attachment for Supplementary figure S2!

Reviewer_1 comments:

Sham Jdeed et al., have performed their study based on TGF beta signaling, and its impact on the EMT process. The study also highlights the mechanism of cancer preventive drugs and the distribution of SWI/SNF complex behind drug treatment. 

The study shows ARID1A orchestration by rexinoid bexarotene and Carvedilol drugs and its effect on EMT-related genes. The following comments might allow the reader to get deeper scientific understanding of the mechanism.

The insights and suggestions from the reviewer are much appreciated. Please find our responses and changes below, which we believe have improved the manuscript.

All implemented changes are marked in the revised manuscript file as ‘Track changes’.

1) From ChIP-seq data it would be nice to show the effect of drugs on genes in cis- and trans- regions specifically related to TGF beta signaling.

The selection of Bex+Carv-regulated genomic regions through SWI/SNF redistribution took place based on two main criteria, the global changes in H3K27 acetylation marks and the functional classification of the putative target genes pointing to phenotype control (EMT) and TGF-b signaling. We recognize that this latter point was not clearly defined in the description of the results and this information has now been introduced in line 274. Thus, because of the genome-wide assessment of acetylation marks in our ChiP-seq data encompasses all cis- AND trans- regions, we posit that the selected genes FoxQ1, KLF4, TGFBR2 and BMP6 among all changes are indeed the genes that could be tied to TGF-b signaling.

2) Comparison of DNA binding and mRNA levels is a nice way to present cis-regulation of transcription factors.

We agree with the reviewer on the importance of this correlation. ARID1A, Brg1 ChIP-qPCR results, as well as the H3K27ac ChIP-Seq data demonstrate the DNA binding events to cis-regulatory regions of target genes. Interestingly, only a slim fraction of select genes fulfilled the requirement of showing a significant change in genomic occupancy by SWI/SNF and altered H3K27 acetylation along with differential transcript levels.

In order to highlight the intercept of these two categories we created a Venn-diagram and introduced it as Supplementary figure S2.

3) Smad4 is shown as an important piece in this mechanism: It was shown that Smad4 levels are disturbed by this mechanism. Smad4 is also known to shift Smad2 and Smad3 complex to nuclear. It would be nice to see if their translocation is also affected.

We studied the nuclear translocation of phosphorylated SMAD2 upon the combined treatment in HME-hTert cells showing a minimal, but not significant decrease in pSMAD2 nuclear fraction. The levels of nuclear phospho-SMAD2 were also suppressed upon ARID1A knock-down (please see attached file for pSMAD2 nuclear part)).

While we believe that this data actually supports our findings (because of the siARID1A effect), due to the lack of significance in the Bex+Carv response we decided not to add this result to the manuscript. However, we added mention of this fact in the results as (‘data not shown’) in line 365 .

4) In turn if, Smad-dependent TGF beta signaling is affected, is there any evidence of a compensation mechanism?

In this study we conducted the genome-wide assessment of H3K27 acetylation and ARID1A/Brg1 enrichment at several genomic sites related to TGF-b signaling and EMT. Our data point to a rather unanimous response of normal breast cells regarding the regulation of cell identity through SMAD transcription factors. One interesting ‘discrepancy’ between SMAD4 and FoxQ1-mediated changes was discovered, in which the cadherins E and N were differentially regulated by TGF-β and Bex+Carv in their response to FoxQ1 or SMAD4, respectively. TGF-β increases the ratio of cells highly ex-pressing N-cadherin, while more of those cells exhibit low nuclear levels of SMAD4. Con-trary to TGF-β, Bex+Carv suppresses N-cadherin expression, along with lower nuclear SMAD4 levels. This has been described in the discussion and might potentially be perceived as a compensatory mechanism.

We hope that these changes will be sufficient for the acceptance of our manuscript.

Reviewer 2 Report

The manuscript is written well. It can be accepted for publication. My only observation is the conclusion (which is written in the last paragraph of the discussion section) should be revised to make it more informative and as a separate paragraph. Some minor changes are required in general English style and flow. 

Author Response

Reviewer_2:

The manuscript is written well. It can be accepted for publication. My only observation is the conclusion (which is written in the last paragraph of the discussion section) should be revised to make it more informative and as a separate paragraph. Some minor changes are required in general English style and flow. 

The insights and suggestions from the reviewer are much appreciated. Please find our responses and changes below, which we believe have improved the manuscript. All implemented changes are marked in the revised manuscript file as ‘Track changes’.

We have revised and reformatted the last segment of the discussion to better highlight the conclusions and examined and corrected the whole text for improvement in its flow. Linguistic errors have also been corrected.

In addition, we created and added a high resolution graphical abstract for a visual summary of the findings (please see the attachment).

Round 2

Reviewer 1 Report

The authors have sufficiently supported the arguments with evidence and addressed all the comments.